# Electric Migration of Hydrogen Ion in Pore-Voltammetry Suppressed by Nafion Film

**Ling Liu [1]**, **Koichi Jeremiah Aoki [2]** and **Jingyuan Chen [1,*]**

[1]   Department of Applied Physics, University of Fukui, 3-9-1 Bunkyo, Fukui 910-0017, Japan;
      liuling@u-fukui.ac.jp
[2]   Electrochemistry Museum, Fukui 910-0804, Japan; kaoki@u-fukui.ac.jp
*   Correspondence: jchen@u-fukui.ac.jp; Tel.: +81-9082625425

**Abstract:** Micro-hole voltammetry exhibiting rectified current-voltage curves was performed in hydrochloric acid by varying the lengths and the diameters of the micro-holes on one end of which a Nafion film was mounted. Some voltammetric properties were compared with those in NaCl solution. The voltammograms were composed of two line-segments, the slope of one segment being larger than the other. They were controlled by electric migration partly because of the linearity of the voltammograms and partly the independence of the scan rates. Since the low conductance which appeared in the current from the hole to the Nafion film was proportional to the cross section area of the hole and the inverse of the length of the hole, it should be controlled by the geometry of the hole. The conductance of the hydrogen ion in the Nafion film was observed to be smaller than that in the bulk, because the transport rate of hydrogen ion by the Grotthuss mechanism was hindered by the destruction of hydrogen bonds in the film. In contrast, the conductance for the current from the Nafion to the hole, enhancing by up to 30 times in magnitude from the opposite current, was controlled by the cell geometry rather than the hole geometry except for very small holes. A reason for the enhancement is a supply of hydrogen ions from the Nafion to increase the concentration in the hole. The concentration of the hydrogen ion was five times smaller than that of sodium ion because of the blocking of transport of the hydrogen ion in the Nafion film. However, the rectification ratio of $H^+$ was twice as large as that of $Na^+$.

**Keywords:** rectified pore-voltammograms of HCl; electric migration; cationic exchange membrane; blocking of ion transport by Nafion; Grotthuss mechanism

## 1. Introduction

Voltammetry through a micro-hole [1–7] has exhibited rectified current-voltage curves, which can be industrially applied to electrochemical sensors [3], to desalination [8–12], and to power generation resulting from salt concentration gradient [13]. A key condition for the rectified voltammograms is asymmetry [6,7,11,14–28] of geometry and functionalization in pores. The asymmetry has been realized by patterning of cylindrical capillaries [14,15,24], chemical modification on walls of holes [16,17], selection of materials of pores [6,18–20], specific voltage control [21], and a nanopipette coated with surfactants [25–28]. A macroscopic concept of the rectification is composed of (i) geometric obstacle of the ion transport through the hole in the closed state and (ii) the localized accumulation of electrolyte owing to ion-exchange membrane in the open state [19,29,30]. A concept from the microscopic viewpoints is a gate linked to the potential distribution in the double-layer or on pore walls [31–37] as well as molecular instability in a hydrodynamic vortex [38,39].

We are interested in a macroscopic explanation of the pore-voltammograms. Pore-voltammograms are composed of two line-segments with different slopes. The linearity of the voltammograms suggests

Ohm's law, or electric migration as a rate-determining step. If the migration occurs in a straight pipe with the cross section area $A$ and length $L$, the current for the mono-valent cation caused by the voltage $V$ between the two ends of the pipe is expressed by [40]

$$I = \lambda c V A / L \tag{1}$$

where $c$ is the concentration of the moving cation, $\lambda \left( = \frac{F^2 D}{RT} \right)$ is the molar conductivity of the ion, and $D$ is the diffusion coefficient. The rectification means that $\lambda c A / L$ should vary with the voltage ($V > 0$ or $V < 0$) [41]. Our aim is to explore the dependence of $\lambda c A / L$ on $V$. The term $\lambda c A / L$ is composed of the geometric variable, $A/L$, and the ionic one, $\lambda c$. The dependence of the conductance on the geometric variable was examined by varying $A$ and $L$ of holes as well as the distance ($L_c$) between two electrodes in the cell [42]. Then, the current of $Na^+$ from the hole to the Nafion was proportional to $A/L$ of the holes, implying the migration-control in the holes. In contrast, the current in the opposite direction was inversely proportional to $L_c$ of the whole cell on the assumption of enhancement of the concentration from the Nafion. The separation of the functions of $\lambda c A / L$ into $A/L$ and $\lambda c$ explained the rectification quantitatively.

The role of $\lambda c$ arises from the interaction of the cation and the cationic exchange membrane. The significance of the interaction can be demonstrated by use of cations other than $Na^+$. A cation distinguished clearly from $Na^+$ is hydrogen ion, in that the diffusion coefficient is ten times larger than that of $Na^+$. We expect a rectification ratio (the ratio of the high conductance to the low one) of hydrogen ion to be ten times larger than that of $Na^+$. The expectation will be examined here by changing $A/L$ of the holes and the cells at some concentrations of HCl for micro-hole voltammograms. We predict that transport of hydrogen ion in a Nafion film is suppressed with highly ionic circumstances of high concentrations of $H^+$ and $-SO_3^-$ in the film which destruct hydrogen bonds in the film, called the Grotthuss effect. The suppressed conductance may be taken as a quantitative degree of the destruction.

## 2. Materials and Methods

A polyethylene terephthalate (PET) film was drilled commercially with a laser beam to 10, 20, 30, and 50 μm in diameter. A tight contact of the PET film with the Nafion film (Sigma-Aldrich) 0.18 mm thick was the key to obtaining not only reproducible voltammograms but also suppression of voltammetric hysteresis. It was challenged by employing an air-pumped pressure or a vice for the two overlapped films, by heating the films up to a temperature less than each melting point, by applying adhesive resign between the two films, and by inserting Nafion-dissolved solution into the hole to evaporate the solvent. The last technique has been used conveniently. However, it was not suitable for our experiments of controlling accurately the Nafion thickness and the length of the hole, because the Nafion solution penetrated in the hole uncontrollably. The successful technique was to apply the pressure with a vice. The double film was bent manually until it was separated owing to the difference in the stresses, as shown in Figure 1C. When bent angles were close to thirty degrees, the reproducible voltammograms were obtained empirically.

The construction of the cell and its magnification are shown in Figure 1A,B, respectively [42]. Two glass pipes with ground glass flanges were faced with each other, sandwiching a Nafion film, a pin-holed (PET) film, and a thin rubber film by means of a clamp for a ball joint. The Nafion was immersed in salt solution for the voltammetric run in one day before voltammetric use. The rubber film avoided leaking of solution.

The working electrode and the counter electrode were platinum coils. Potential was measured with a sense electrode and a reference electrode, both of which were silver wires covered with AgCl as a local sensor of voltage. Only a small part on the wire tip was exposed to a solution by coating AgCl with resign. The current was supplied between the working electrode and the counter electrode, whereas the voltage was obtained with the sense electrode and the reference electrode by means of

the four-electrode system by use of a potentiostat, Compactstat (Ivium, Eindhoven, The Netherlands), at 20 ± 2 °C.

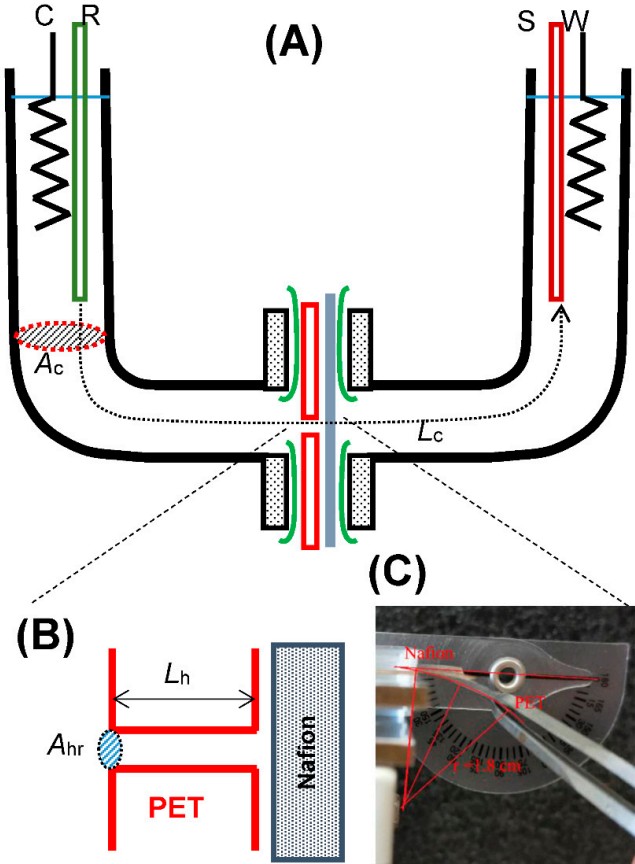

**Figure 1.** Illustrations of the structure of the cell (**A**), the magnified hole (**B**), and (**C**) photograph of the artificially bent double film composed of the PET and the Nafion film pressed with a vice.

In order to estimate the conduction of the Nafion film, we measured voltammograms of the cell separated with and without the Nafion film in 1 M HCl solution without the PET film. Both the voltammograms observed with/without Nafion film were almost identical with only 2% difference in current, indicating that the Nafion film should have a negligibly small effect on the overall resistivity for pore voltammograms.

## 3. Results and Discussion

Figure 2 shows hole-voltammograms of 0.1 mM HCl solution (a) with and (b) without the Nafion film. Since the $H^+$-typed Nafion had 0.18 Ω m in Section 2, the resistance near the hole with the geometric parameter $A_h/L_h = 50$ μm is 3.6 kΩ, where $A_h$ is the cross section area ($=\pi r_h^2$ for the radius $r_h$ of the hole) and $L_h$ is the length of the hole. This causes 14 mV voltage shift in voltammograms at our maximum current 4 μA (Figure 2). The voltammograms had less hysteresis with a decrease in the scan rates. The voltammograms showed a decrease in hysteresis with a decrease in the scan rates. When we take a degree of the hysteresis as the ratio of the difference between the largest and the smallest currents to the middle current at $V = 1.5$ V, the degrees were 0.30, 0.18, and 0.09 for $v = 0.10$, 0.05, and 0.01 V·s$^{-1}$. The voltammogram without Nafion (b) exhibited a line through the origin—i.e., obeying Ohm's law. In contrast, the Nafion film caused the rectified current-voltage curve, of which peculiarity was revealed in the slope for $V > 0$. The linearity in domains for $V > 0$ or $V < 0$ also supports Ohm's law. The former conductance was 30 times larger than the latter.

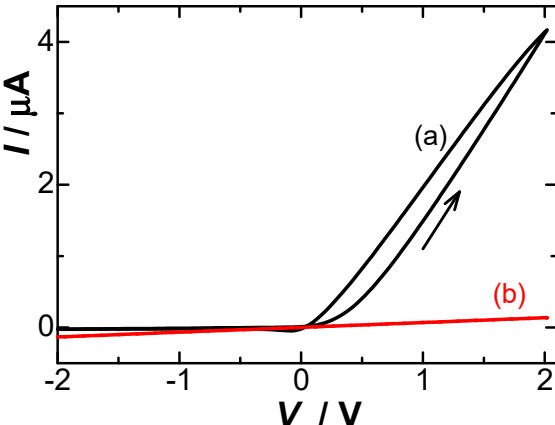

**Figure 2.** Typical pore-voltammograms in 0.1 mM HCl solution (a) with and (b) without the Nafion film at the scan rate 0.1 V·s$^{-1}$. The hole geometry was 50 μm long and 50 μm in diameter. The arrow means the direction of the voltage scan.

First, we examined variations of the conductance with $A_h/L_h$ by altering the diameters of the holes and thickness of the PET film when no Nafion film was equipped. Figure 3a,c show the dependence of the conductance $G$ obtained from the slope of the voltammogram on $A_h/L_h$. The linearity for both (a) HCl and (b) NaCl solutions indicates that the hole works as a cylindrical, homogeneous conductance, which can be expressed by

$$G = (A_h/L_h)(\lambda_+ + \lambda_-) \tag{2}$$

where $c_b$ is concentration of HCl or NaCl in the bulk, and $\lambda_+$ and $\lambda_-$ are ionic molar conductivities of the cation (H$^+$ or Na$^+$) and the anion (Cl$^-$), respectively. The slope of the line for HCl was larger than that for NaCl because of $\lambda_H > \lambda_{Na}$. The conductance of HCl is predicted to be caused both by H$^+$ and Cl$^-$, whereas that of NaCl is by Na$^+$ and Cl$^-$. By use of bibliographic values of $\lambda_H = 35 \times 10^{-3}$ S m$^2$ mol$^{-1}$, $\lambda_{Na} = 5.0 \times 10^{-3}$ S m$^2$ mol$^{-1}$, and $\lambda_{cl} = 7.5 \times 10^{-3}$ S m$^2$ mol$^{-1}$, we obtain the ratio

$$r_G \equiv \frac{G_{HCl}}{G_{NaCl}} = \frac{\lambda_H + \lambda_{Cl}}{\lambda_{Na} + \lambda_{Cl}} = 3.4 \tag{3}$$

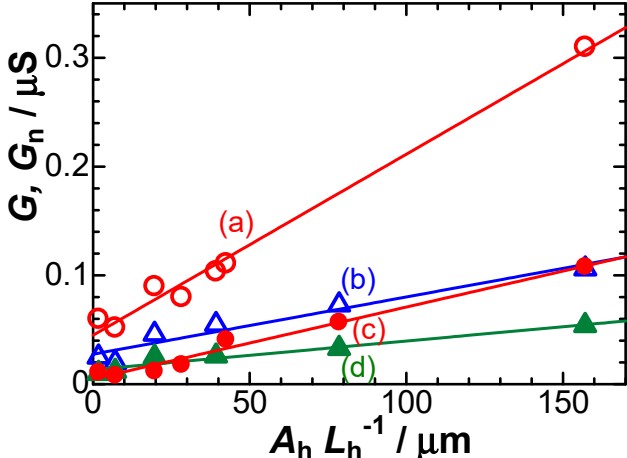

**Figure 3.** Variations of the conductance in (a, c) 0.1 mM HCl and (b, d) 0.1 mM NaCl solution with the hole dimension, $A_h/L_h$, in (a, c) the absence and (b, d) presence of the Nafion film.

This is the same as the experimental ratio of the slopes in Figure 3a,c. The intercept in Figure 3 can be attributed to leakage of penetrated ions through the ground glass flanges at the edge of the cell [42].

Next, the mounting of the Nafion film bent the linear voltammograms at $V = 0$ drastically to exhibit two line-segments, as shown in Figure 2. It largely enhanced the slope for $V > 0$, whereas it slightly decreased the slope for $V < 0$. Figure 3b,d show the dependence of the conductance $G_n$ for $V < 0$ on $A_h/L_h$. The Nafion film decreased $G_n$ from $G$ without the Nafion film (a, c), common to HCl and NaCl solutions. The decrease in $G_n$ can be readily explained by the hindrance of the charge-carrying anion (Cl⁻) owing to the immobilized sulfonic moiety of Nafion. Since the hindrance is represented quantitatively by the replacement of $\lambda_+ + \lambda_-$ by $\lambda_+$, the ratio becomes $r_G = \lambda_H/\lambda_{Na}$, of which the bibliographical value is 7.0. The experimental value from Figure 3 (the ratio of the slope of (d) to (c)) was 2.5. The difference in the values means the transport of hydrogen ion is hindered with the Nafion film, or that its rate of hydrogen ion in the Nafion is smaller than that in the solution. When we replace $\lambda_H$ in the bulk by $\lambda_{H-Nafion}$ in the Nafion—i.e., $\lambda_{H-Nafion}/\lambda_{HNa} = 2.5$—we can evaluate $\lambda_{H-Nafion} = 13 \times 10^{-3}$ S m² mol⁻¹. This value is rewritten to the diffusion coefficient $D = 3.4 \times 10^{-5}$ cm² s⁻¹ through $\lambda = F^2 D/RT$. This effect can be explained in terms of hindrance of the Grotthuss mechanism [43–45], in which hydrogen ion diffuses through the hydrogen bond network via formation and cleavage of covalent bonds. The highly ionic atmosphere in Nafion film destructs hydrogen bonds to decrease the extraordinarily large value of the diffusion coefficient of hydrogen ion.

We examined quantitatively a degree of the Grotthuss effect by determining diffusion coefficients of hydrogen ion in various concentrations of NaCl solution including 1 mM HCl by cyclic voltammetry. The cathodic peak for the reduction of H⁺ was observed at −0.6 V vs. Ag|AgCl in cyclic voltammograms at the platinum disk electrode 1.6 mm in diameter. The peak currents were proportional to the square-roots of the scan rates, $v$, less than 0.1 V s⁻¹. Unfortunately, the diffusion coefficient cannot be determined unambiguously from the slope of the proportionality because the slope depends on whether the reduction obeys sequential or concomitant mechanisms of the two-electron reduction [46,47]. We assume that the reduction current density obeys the sequentially one-electron transfer through $j_P = 0.446\, c_b F(v D_{ap} F/RT)^{1/2}$ for the apparent diffusion coefficient $D_{ap}$. Since we are interested in only the slope of $I_p$ vs. $v$, any mechanism is satisfied. Values of $D_{ap}$ were plotted against concentration of NaCl in Figure 4. They were kept constant for [NaCl] ≤ 1.4 M but decreased at the higher concentrations. This is the hindrance of the Grotthuss effect. The variation in Figure 3 demonstrated that the $D$-value in the Nafion film decreased by one third from the bulk. Comparing this fact with Figure 4 indicates that salt effect in the Nafion film should be stronger than that in 6 M in bulk.

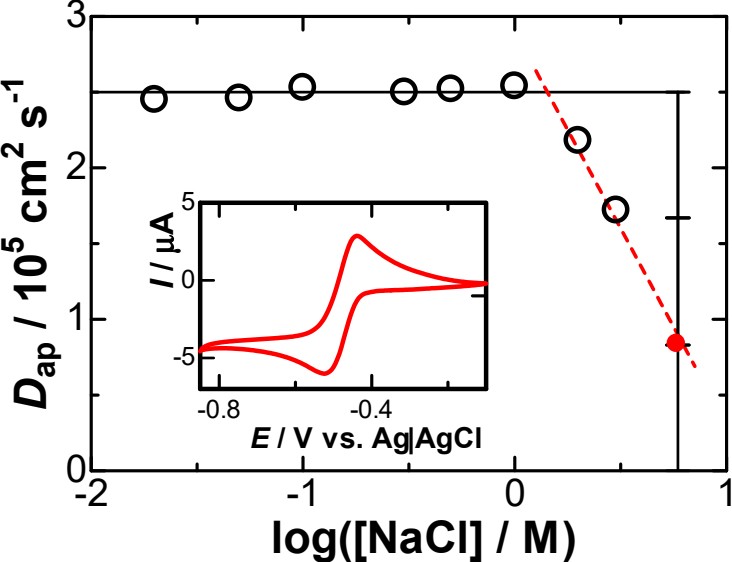

**Figure 4.** Dependence of the apparent diffusion coefficient of hydrogen ion obtained from the cathodic peak currents by cyclic voltammetry at $c_b = 1$ mM on concentrations of NaCl solutions. The inset is the cyclic voltammogram of 1 mM HCl at $v = 0.01$ V s⁻¹.

The interesting voltammetric behavior lies in line-segment for $V > 0$ of which conductance, $G_p$, is much higher than $G_n$ for $V < 0$. Figure 5 shows plots of $G_p$ against $A_h/L_h$ in the common concentration of HCl and NaCl solutions. The values of $G_p$ for $A_h/L_h < 40$ μm are almost proportional to $A_h/L_h$, whereas those for $A_h/L_h > 40$ μm tend to a constant. These variations of HCl are similar to those of NaCl. The proportionality for $A_h/L_h < 40$ μm indicates that the conductance should be controlled by the hole geometry, as is for $G_n$. Since transport of Cl$^-$ is blocked with the immobilized sulphonate ions of the Nafion film [48], the conductance is provided by only the cation in the hole—i.e., $\lambda_+ c_+$—where $c_+$ is the concentration of the cation in the hole. The ratio of the slope for HCl to that for NaCl is expressed by

$$r_G = G_{p,HCl}/G_{p,NaCl} = \lambda_H c_H/\lambda_{Na} c_{Na} \tag{4}$$

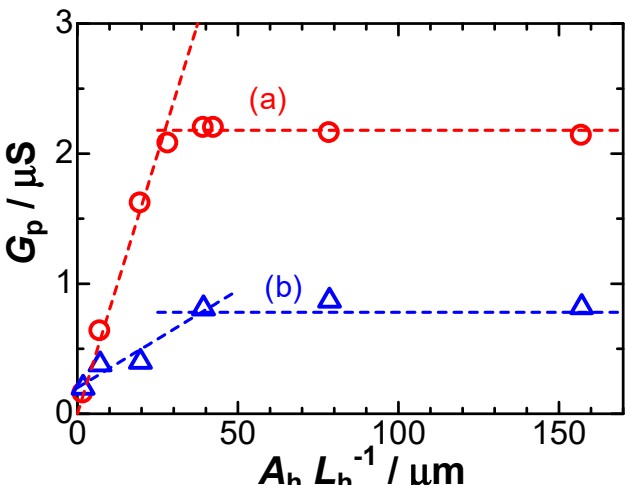

**Figure 5.** Variations of the conductance, $G_p$, for $V > 0$ with the hole dimension, $A_h/L_h$, in (a) 0.1 mM HCl and (b) 0.1 mM NaCl solution.

If $c_H$ and $c_{Na}$ take a common value, $r_G$ becomes $\lambda_H/\lambda_{Na} = 7.0$ bibliographically. However, the experimental ratio was ca. 5 from Figure 5 in spite of some ambiguity of the plot. The low experimental value can be explained in terms of the hindrance of transportation of the hydrogen ion in the Nafion film. However, it is not 2.5 for $G_n$ in Figure 3. There is complication in alteration of $c_H$, discussed later.

Constancy of $G_p$ for $A_h/L_h > 40$ μm in Figure 5 does not specify any controlling step by the geometry. According to the previous report [42], values of $G_p$ of NaCl have been controlled by the migration through the glass cell rather than the hole in which the cation concentration is enhanced by a supply from the Nafion film. In order to examine the migration control in the glass cell other than the hole, we obtained voltammograms for some distances between two Ag|AgCl electrodes for the given hole geometry in the constant domain of $G_p$ in Figure 5. Figure 6 shows variations of the cell resistance ($1/G_p$) with the cell geometric parameter, $L_c/A_c$, where $L_c$ is the distance between the two voltage-sensing electrodes and $A_c$ is the cross section area of the glass cell (see Figure 1A of [43]). The plots show a linear variation, of which intercept can be assigned to resistances of the hole and the Nafion film in series. Thus, the total resistances without and with Nafion are expressed as

$$(G_p^{-1})_{noNaf} = (L_h/A_h)/c_b(\lambda_+ + \lambda_-) + (L_c/A_c)/c_b(\lambda_+ + \lambda_-) \tag{5}$$

$$(G_p^{-1})_{Naf} = (L_h/A_h)/c_+\lambda_+ + (L_c/A_c)/c_b(\lambda_+ + \lambda_-) \tag{6}$$

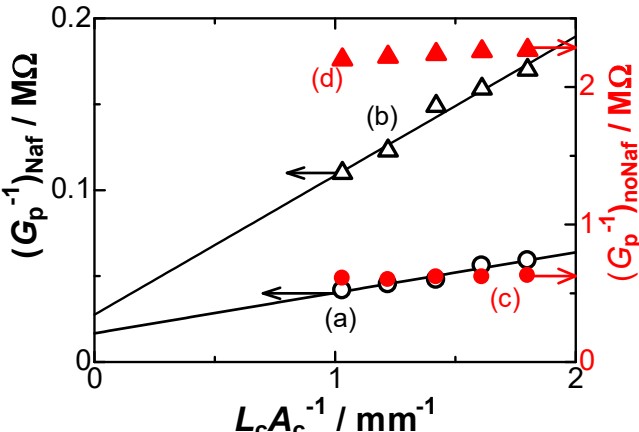

**Figure 6.** Variations of $1/G_p$ with $L_c/A_c$ for (a, c) HCl and (b, d) NaCl solution of $c_b$ = 0.1 mM with (a, b) and without (c, d) the Nafion film in the hole 0.05 mm in diameter and 0.05 mm in length.

The values of the slopes for HCl (a, c) were almost common (23 Ωm) (the right scale being different from the left one), and those for NaCl (b, d) were also common (82 Ωm). The theoretical values of $1/c_b(\lambda_+ + \lambda_-)$ for HCl and NaCl are, respectively, 24 and 80 Ωm. The agreement of the experimental values with the theoretical ones indicates that the carrier of the current should be combinations of {H$^+$ and Cl$^-$} and {Na$^+$ and Cl$^-$}. The ratios of the intercepts were 35 for HCl and 79 for NaCl, which equal $c_+\lambda_+/\{c_b(\lambda_+ + \lambda_-)\}$, according to Equations (5) and (6). Then, we have $c_+$ = 0.043 M for HCl and $c_+$ = 0.2 M for NaCl at $c_b$ = 1 mM. The enhancement of $c_+$ from $c_b$ is caused by the supply of Na$^+$ and H$^+$ from the Nafion film by the cationic flow by the cell-geometric migration. The amount of the supply of H$^+$ is the one-fifth of that of Na$^+$ because of the hindrance of the Grotthuss mechanism.

One question is whether the evaluation of $c_H/c_{Na}$ = 0.043/0.2 = 0.21 at the mouth of the hole on the Nafion side is valid only for the concentration ($c_b$ = 1 mM) in Figure 6 or not. It can be solved by examining concentration dependence of $G_p$. We examined the dependence of $G_p$ on $c_b$, and plotted the resistance ($1/G_p$) against the concentrations in Figure 7 for some combinations of hole geometry. Most points except for the lowest concentration fell on a line. As a result, the concentration of H$^+$ in the hole suppled from the Nafion is proportional to $c_b$, or $c_H/c_{Na}$ = 0.21 valid for 0.1 mM < $c_b$ < 3 mM. The proportionality has been recognized in other researchers' data [49], although a small value of an intercept was found.

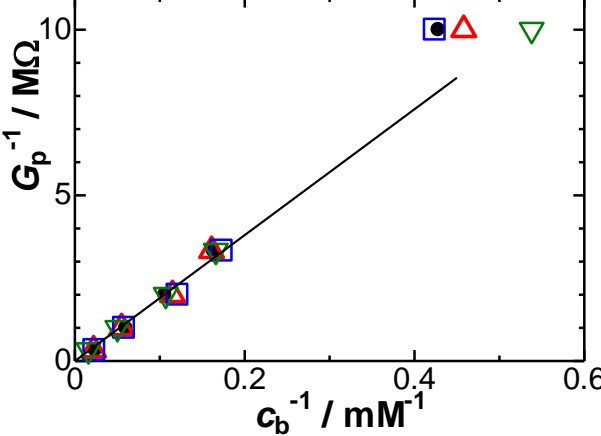

**Figure 7.** Dependence of the resistance for $V$ > 0 on the inverse of the bulk concentration at hole geometry with $A_h/L_h$ = (filled circles) 42.5, (triangles) 19.6, (squares) 79.5, and (inverse triangles) 157 μm in the range from $c_b$ = 0.1 to 3 mM.

Another question is which out of $Na^+$ or $H^+$ can provide a larger rectification ratio ($G_p/G_n$). The hindrance of the transport of $H^+$ affects both $G_p$ through $c_H = 0.2c_{Na} >> c_b$ at the mouth of the hole near the Nafion and $G_n$ through $\lambda_{H\text{-Nafion}} = \lambda_H/3$ in the Nafion. This relation seems to provide $(G_p/G_n)_H < (G_p/G_n)_{Na}$. Our results—$r_G = 2.5$ for $V < 0$ and $r_G = 5.0$ for $V > 0$—indicate that $(G_p/G_n)_H = 2(G_p/G_n)_{Na}$. The twice as large rectification ratio is provided by the transport rate of $H^+$ in the hole ten times larger than that of $Na^+$.

## 4. Conclusions

Our method of examining the hole-voltammograms is to discuss the parameter, $\lambda_+c_+A/L$, of the electric migration. We compare the hole-voltammetric conductance of HCl with that of NaCl, and take the ratio as $r_G = G_{HCl}/G_{NaCl}$. The ratio without Nafion is $r_G = 3.4 = (\lambda_H + \lambda_{Cl})/(\lambda_{Na} + \lambda_{Cl})$ owing to the participation both in the cations and $Cl^-$. This value agrees with the bibliographically calculated value. The Nafion film divides a linear voltammogram into two line-segments divided at $V = 0$. $G_n$ for $V < 0$, the current flowing from the hole to the Nafion, is controlled by the hole geometry $A_h/L_h$ as well as $\lambda_+c_b$. $Cl^-$ does not contribute to the conductance because of blocking of $Cl^-$ by the immobilized $-SO_3^-$. Then, the ratio is bibliographically $r_G = \lambda_H/\lambda_{Na} = 7.0$, whereas the experimental value is 2.5 owing to the suppression of transport of $H^+$ in the Nafion film by the destruction of hydrogen bond or hindrance of the Grotthuss mechanism.

$G_p$ for $V > 0$ in the opposite direction of the current is controlled by the cell geometry $A_c/L_c$ as well as $\lambda_+c_+$ in the hole. The concentration in the hole, $c_+$, is supplied from the Nafion film so that 0.2 M for $Na^+$ and 0.043 M for $H^+$. The lower concentration of $H^+$ is ascribed to the hindrance of the Grotthuss mechanism. The rectification ratio ($G_p/G_n$) of $H^+$ is twice as large as that of $Na^+$ because the transport rate of $H^+$ in the hole is ten times larger than that of $Na^+$.

**Author Contributions:** Conceptualization, K.J.A.; Methodology, L.L.; Validation, J.C., K.J.A.; Formal analysis, L.L.; Investigation, L.L.; Resource, J.C.; Writing-original draft preparation, L.L.; Writing-review and editing, K.J.A.; Visualization, L.L.; Supervision, J.C.; Project administration, J.C.; Funding acquisition, J.C.; All authors contributed equally to this manuscript. All authors have read and agreed to the published version of the manuscript.

**Funding:** This research received no external funding.

**Conflicts of Interest:** The authors declare no conflict of interest.

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
