# Peer review of "Electric Migration of Hydrogen Ion in Pore-Voltammetry Suppressed by Nafion Film"

_2673-3293, doi:10.3390/electrochem1040027_

Round 1
Reviewer 1 Report
Comments in the attached file

Reviewer 2 Report
The manuscript describes the dependence of the hole and cell geometries on the conductance of an experimental micro hole voltammetry system consisting of PET and Nafion films. The introduction section is rather concise, but sufficient citations are given and will be useful to the readers. The experimental data are clearly presented. However, I have some parts that I am concerned about. The reason for this and the points to be considered are described briefly as follows.
(1) The ionic conductivity of Nafion is strongly dependent on temperature and humidity. It must be said that the accuracy of the data is questionable from a reliability standpoint, since no information on temperature and humidity is described in the experimental section as well as throughout the paper. Is it possible to provide information on temperature and humidity? Or is it not controlled for temperature and humidity in the first place?
(2) With respect to Figure 4, it is preferable that the original CV is shown. I would suggest that it could be shown in the multiples along with the original figure 4, or added as supplementary material.
(3) The authors mention the Cl- would not contribute to the conductance because it is blocked by the immobilized SO3- group without citation in lines 173-174. Some readers may find this statement ambiguous, please cite the following paper to avoid it. (E. K. Unnikrishnan et al., J. Membr. Sci. 1997, 137, 133-137.)
In summary, the paper is recommended for publication in Electrochem after minor revisions.
Author Response
The manuscript describes the dependence of the hole and cell geometries on the conductance of an experimental micro hole voltammetry system consisting of PET and Nafion films. The introduction section is rather concise, but sufficient citations are given and will be useful to the readers. The experimental data are clearly presented. However, I have some parts that I am concerned about. The reason for this and the points to be considered are described briefly as follows.
(1) The ionic conductivity of Nafion is strongly dependent on temperature and humidity. It must be said that the accuracy of the data is questionable from a reliability standpoint, since no information on temperature and humidity is described in the experimental section as well as throughout the paper. Is it possible to provide information on temperature and humidity? Or is it not controlled for temperature and humidity in the first place?
Answer:
We added the temperature, 20±2 oC, to the end of the third paragraph.
We have the data of conductivity of a wet Nafion film, which are more concise than the evaluation of conductance of the strip film. We replaced the forth paragraph by
" In order to estimate the conduction of the Nafion film, we measured voltammograms of the cell separated with and without the Nafion film in 1 M HCl solution without the PET film. Both the voltammograms observed with/without Nafion film were almost identical with only 2 % difference in current, indicating that the Nafion film should have a negligibly small effect on the overall resistivity for pore voltammograms."
(2) With respect to Figure 4, it is preferable that the original CV is shown. I would suggest that it could be shown in the multiples along with the original figure 4, or added as supplementary material.
Answer:
We inserted a voltammogram of HCl as an inset in Fig. 4.
(3) The authors mention the Cl- would not contribute to the conductance because it is blocked by the immobilized SO3- group without citation in lines 173-174. Some readers may find this statement ambiguous, please cite the following paper to avoid it. (E. K. Unnikrishnan et al., J. Membr. Sci. 1997)
Answer:
The suggestion is quite reasonable. We added reference [49] to the manuscript.
